



# Stationary Waves and Upward Troposphere-Stratosphere Coupling in S2S Models

Chen Schwartz*[1], Chaim I. Garfinkel*[1], Priyanka Yadav[2], Wen Chen[4,5], and Daniela Domeisen[2,3]

[1]The Fredy and Nadine Herrmann Institute of Earth Sciences, Hebrew University of Jerusalem, Jerusalem, Israel
[2]Institute for Atmospheric and Climate Science, ETH Zurich, Zurich, Switzerland
[3]Institute of Earth Surface Dynamics, University of Lausanne, Lausanne, Switzerland
[4]College of Earth and Planetary Sciences, University of Chinese Academy of Sciences, Beijing, China
[5]Institute of Atmospheric Physics, Chinese Academy of Sciences, Beijing, China

**Correspondence:** Chen Schwartz (chen.schwartz1@mail.huji.ac.il), Chaim I. Garfinkel (chaim.garfinkel@mail.huji.ac.il)

**Abstract.** The simulated Northern Hemisphere stationary wave (SW) field is investigated in 11 subseasonal-to-seasonal (S2S) models. It is shown that while most models considered can well-simulate the stationary wavenumbers 1 and 2 during the first two weeks of integration, they diverge from observations following week 3. Those models with a poor resolution in the stratosphere struggle to simulate the waves, both in the troposphere and the stratosphere, even during the first two weeks, and biases extend from the troposphere all the way up to the stratosphere. Focusing on the tropospheric regions where SWs peak in amplitude reveals that the models generally do a better job in simulating the Northwest Pacific stationary trough, while certain models struggle to simulate the stationary ridges both in Western North America and the North Atlantic. In addition, a strong relationship is found between regional biases in the stationary height field and model errors in simulated upward propagation of planetary waves into the stratosphere. In the stratosphere, biases mostly are in wave-2 in those models with high stratospheric resolution, whereas in those models with low resolution in the stratosphere, a wave-1 bias is evident, which leads to a strong bias in the stratospheric mean zonal circulation due to the predominance of wave-1 there. Finally, biases in both amplitude and location of mean tropical convection and the subsequent subtropical downwelling, are identified as possible contributors to biases in the regional SW field in the troposphere.

## 1 Introduction

The Northern Hemisphere (NH) climate is not uniform in longitude, despite the incoming solar radiation being roughly zonally symmetric on a daily average. This zonal asymmetry in the NH climate is a result of large-scale asymmetries in the lower boundary of Earth, that force large-scale waves that are stationary in nature, and are stronger in amplitude during boreal winter compared to boreal summer (Held et al., 2002; Garfinkel et al., 2020).

Previous works have found that large-scale orography, such as the Rockies in North America and the Tibetean Plateau in Asia, plays a major role in forcing the NH stationary waves (SWs hereafter). Land-sea contrast, driven by differences in heat capac-

---

*these authors contributed equally





ity and friction between ocean and continents, also is a crucial driver of stationary waves, with the relative importance of the two dependent on the region and also subject to nonlinear interactions (Garfinkel et al., 2020). In addition, zonal sea-surface temperature (SST) and diabatic heating anomalies in extratropical and tropical regions, have been found to be a major driver

of the large-scale extratropical SWs pattern and circulation (Held et al., 2002). Tropical mid-tropospheric diabatic heating is balanced by adiabatic cooling, while extratropical mid-tropospheric diabatic heating is balanced by a large-scale meridional circulation (Hoskins and Karoly, 1981). Hence both tropical and extratropical diabatic heating can act as a Rossby wave source (RWS), as both lead to the advection of barotropic absolute vorticity by the upper tropospheric divergent flow (Sardeshmukh and Hoskins, 1988).


Realistic representation of extratropical SWs is highly important for general circulation models, as they influence weather and climate over densely populated areas in Europe, North America and Asia. SWs can modulate the trajectories of mid-latitude storms and play a major role in shaping the distribution of surface temperatures and moisture along comparable latitude bands (Simpson et al., 2016). Therefore, even small biases in models' representation of these waves and their drivers, can lead to

significant errors in regional weather and climate forecasts and projections (Neelin et al., 2013). In particular, the role played by regional diabatic heating biases in generating biases in extratropical SWs in the Coupled Model Intercomparison Project Phase 5 (CMIP5) models has been recently demonstrated by Park and Lee (2021). Specifically, they show that biases in large-scale tropical convection, both over the Western Pacific and Western Atlantic regions, have direct and indirect impacts on SWs biases over the North Pacific and North Atlantic sectors. Their findings highlight the importance of well-represented time-mean

tropical convection in climate models for reliable regional climate projections over the extratropics.

SWs play a critical role in troposphere-stratosphere upward coupling. In the NH, SWs forced mostly by topography extend from the troposphere up to the stratosphere and weaken the mid-winter polar vortex (Charney and Drazin, 1961; Garfinkel et al., 2020). Furthermore, transient wave activity in phase with the SWs (e.g., ridges of the transient waves are collocated with

those of the SWs) leads to transient weakening of the vortex (Garfinkel and Hartmann, 2008; Garfinkel et al., 2010; Cohen and Jones, 2011; Domeisen and Plumb, 2012; Smith and Kushner, 2012; Watt-Meyer and Kushner, 2015), and in extreme cases, breakdown of the vortex during a sudden stratospheric warming event, with the phase speed of the transients playing a role before those extreme events (Domeisen et al., 2018; Baldwin et al., 2021). Instead, the polar vortex strengthens during periods of anomalously low wave activity in the troposphere and stratosphere (Limpasuvan et al., 2005). Therefore, the accuracy of

models in simulating the amplitude and longitudinal phase of the SWs is crucial for capturing the wintertime stratospheric mean state, which has direct implication on variability that can potentially enhance surface weather predictability on subseasonal timescales (Kolstad et al., 2010; Kidston et al., 2015; Domeisen et al., 2020b).

The Subseasonal-to-Seasonal (S2S) Prediction project (Vitart et al., 2017) has recently made available a large number of

hindcasts covering the past several decades. These simulations are all initialized with observed sea surface temperatures and the atmospheric conditions, and as they are intended to be useful for forecasting operationally, they can be compared directly



to observed variability during the duration of their forecast. Previous works have used these models to assess their skill in capturing both tropospheric and stratospheric teleconnections that lead to polar vortex variability (Schwartz and Garfinkel, 2020; Garfinkel et al., 2018, 2019). Specifically, Schwartz and Garfinkel (2020) investigated biases in SWs in 5 subseasonal forecast models to examine their SW pattern biases and their implications for upward coupling resulting from intraseasonal tropical variability in the troposphere. They found that the NCEP, ECMWF and UKMO models have realistic SW patterns, particularly during the first week of reforecast, while biases in the CMA and BoM models are more pronounced throughout the run. They also demonstrate the importance of realistic simulated SWs for upward coupling, with the models with better simulated SWs, in particular over the North Pacific (NP) region, also simulating a more realistic upward coupling in response to subseasonal tropical variability.

In this work, we examine the fidelity of the simulated SWs both in the troposphere and stratosphere, and the implications for upward coupling and the stratospheric mean state, as represented in 11 subseasonal forecast models, and for three models we consider two distinct versions. Furthermore, we investigate the tropical biases that potentially contribute to the extratropical SWs biases in these models.

This paper is organized as follows. The data and methods used for the analysis are described in section 2. In section 3, the fidelity of the Northern Hemisphere tropospheric SWs is examined in subseasonal forecast models, with an emphasis put on regions where SWs are of large amplitude. In section 4, the SWs biases in the models are discussed in the context of troposphere-stratosphere coupling and the simulated mean stratospheric circulation. In section 5, the possibility that biases in regional SWs in the extratropics emanate from model biases in the distribution of tropical convection is considered. Finally, conclusions and a discussion are in section 6.

## 2  Data and Methods

The fidelity of the stationary waves is examined in models that have contributed to the S2S Prediction project (Vitart et al., 2017). We include all eleven modeling centers - the Australian Bureau of Meteorology (BoM), the European Centre for Medium-Range Weather Forecasts (ECMWF), the China Meteorological Administration (CMA), the United Kingdom Met. Office (UKMO), the National Center for Environmental Prediction (NCEP), the Korean Meteorological Administration (KMA), Japan Meteorological Agency (JMA), The Institute of Atmospheric Sciences and Climate of the National Research Council of Italy (ISAC-CNR), Hydrometeorological Centre of Russia (HMCR), Environment and Climate Change Canada (ECCC) and Meteo France (CNRM). A summary of the reforecast availability, ensemble size, and selected properties of the vertical resolution is presented in table 1. Throughout the paper, we look at the ensemble mean rather than individual ensemble members. For the UKMO, we downloaded hindcasts for the operational model in use during 2015 and the winter of 2019/2020, for the ECMWF, we downloaded data for the model version in use during 2016 and the winter of 2019/2020 (CY41R1/CY41R2 and CY46R1), and for the CNRM we downloaded data of model versions 2014 and 2019. For the ECMWF model, we use only





one reforecast each week, and for the NCEP model we only downloaded 9 reforecasts each month, for consistency with the
data availability for the other models. These various models differ in the quality of their representation of the stratosphere
(Domeisen et al., 2020a): the stratosphere is less well resolved in CMA, CNR-ISAC, HMCR, and BoM as compared to the
other models (Table 1). We consider reforecasts initialized in November through February and assess the stationary waves as
a function of forecasted week. Note that each modeling center has made available reforecasts from different years and the

reforecast initialization dates differ among the models even for a given year.

We define the stationary waves by first computing the time mean geopotential height over intializations during November-
December-January-February (NDJF) for each model and forecast week, and then subtract off the zonal mean height at each
latitude. For meridional eddy heat flux, we calculate the anomalies from the zonal mean of daily meridional wind, v, and tem-
perature, T, separately, then multiply them, and average over all initializations in NDJF to obtain $\overline{v^* T^*}$, where overbar denotes

the zonal mean and asterisk denotes deviations from the zonal-mean. We do not explicitly filter out short-time variability, al-
though these waves often take non-stationary features, however, by averaging over many initializations and over week-periods,
the non-stationary features are filtered out. The wavenumber components of the geopotential height and meridional heat flux
fields are obtained by performing a Fourier decomposition and cross-spectrum respectively. Each field is averaged over week-
periods, thus we look at weekly time leads from initialization.

Diabatic heating is not available as a standard output in the S2S archive. The Lagrangian pressure tendency (i.e. vertical veloc-
ity on pressure coordinates, or $\omega$) is available for all models, and we use $\omega$ at 500hPa as a proxy for convection. Most models
make available outgoing longwave radiation (OLR), and the biases in OLR resemble those for $\omega$.



Table 1: S2S Model experiments chosen

| model (ensemble members) | years | reforecasts analyzed | vertical levels | model-top |
|---|---|---|---|---|
| CMA (BCC-CPS-S2Sv1) (4) | 1999-2014 | 6 per month | 40 | 0.5hPa |
| NCEP (CFSv2) (4) | 1999-2010 | 9 per month | 64 | 0.02hPa |
| ECMWF 2016 (CY41R1/CY41R2/CY43R1) (11) | 1996-2013 | 4-5 per month | 91 | 0.01hPa |
| ECMWF 2019/2020 (CY46R1) (11) | 1999-2019 | 4-5 per month | 91 | 0.01hPa |
| BoM (POAMA P24) (33) | 1981-2013 | 6 per month | 17 | 10hPa |
| UKMO 2015 (GloSea5) (3) | 1998-2009 | 4 per month | 85 | 85km |
| UKMO 2020 (GloSea5-GC2-LI) (7) | 1993-2016 | 4 per month | 85 | 85km |
| KMA (GloSea5-GC2) (3) | 1991-2016 | 4 per month | 85 | 85km |
| Météo France-2019 - (CNRM-CM 6.1) (10) | 1993-2017 | 3-5 per month | 91 | 0.01hPa |
| Météo France 2014 (CNRM-CM 6.1) (4) | 1993-2014 | 4 per month | 91 | 0.01hPa |
| JMA (GEPS1701) (5) | 1981-2012 | 2 per month | 60 | 0.01hPa |
| CNR-ISAC (GLOBO) (5) | 1981-2010 | 5-7 per month | 54 | 6.8hPa |
| HMCR (RUMS) (10) | 1985-2010 | 4-5 per month | 28 | 5hPa |
| ECCC (GEPS 6) (4) | 1998-2017 | 4-5 per month | 45 | 0.1hPa |

## 3 Fidelity of Tropospheric Stationary Waves in Subseasonal Models

We begin our analysis with the fidelity of the full mid-tropospheric stationary wave field in the models. Figure 1 shows the
500 hPa eddy geopotential height field in the NCEP model, the corresponding days in ERA-I during weeks 1, 3, 5 and 6 of integration, and the model biases.

During the first week, biases in the stationary wave field are small over the mid-high latitudes. However, during week 3, biases are already developed, with a strong positive and negative biases over the Northwest and Northeast Pacific, respectively. In addition, a negative bias develops over the Northwestern Atlantic. Following week 3, biases in the North Pacific sector strengthen
in magnitude, while those over the North Atlantic slightly strengthen in magnitude and strongly project onto the negative node of wave-2 over the Northwestern Atlantic (see magenta contours).

The biases in week 3 are summarized in Figure 2. The green line corresponds to NCEP, and the other models are shown with other colors. Figure 2a shows a longitudinal cross-section of 500 hPa eddy height field at 50°N during week 3 both for the models (solid line) and ERA-I (dashed line), and figure 2b shows the biases of the models. In week 3, biases are evident par-





ticularly in the ridges over Western North America and Northern Europe/Atlantic, and in the Northwest Pacific trough. The largest biases are simulated by the ISAC model in Western North America and by HCMR, CMA and BoM over the North Atlantic (figure 2b). In the Northwest Pacific sector, the biases are relatively small, with the BoM showing the largest bias over this region.

As seen in figure 2, the mid-tropospheric SW biases in the models, especially those which perform poorly, are mostly con-

fined to three key regions in which the observed SW amplitude is relatively large. Those regions are the Northwest Pacific (160°E-170°E), Western North America (230°E-240°E) and North Atlantic (345°E-355°E) sectors. Figure 3a-c demonstrates the development of weekly lead-time SWs biases, with a mid-latitude (45°N-55°N) cross-section of 500 hPa stationary eddy height in these three key regions, respectively, both in the models (solid lines) for initializations during NDJF and the corresponding days in observations (dots).

In the Northwest Pacific, about half of the models are already biased one week after initialization, and the bias persists throughout the run (3a). However, the bias is relatively small. The BoM model, which has the largest bias, reaches a maximal error of approximately 10% in week 2, while other models have even a smaller bias. Note the ECMWF-2019 model (thin black line) which simulates a too shallow trough in the NW Pacific at all lead times, while during the first four weeks the observed trough deepens, resulting in an increase of the bias. The CMA, JMA, HMCR, BoM and the ISAC model are biased from week 1.

In contrast, The UKMO-2019, CNRM-2014 and KMA simulate biases of less than 5% of the NW Pacific trough throughout the run, while the ECCC model is the only model that almost accurately simulates the trough during all weeks. Other models simulate imperceptible biases for the NW Pacific trough in week 1, but afterwards biases grow.

While the biases in the Northwest Pacific are less than 10% of the observed trough for all models, biases are relatively larger in the other regions. In Western North America, the spread among models is relatively small by the end of week 1, although

the bias of the simulated predominant stationary ridge over this region becomes larger at later lags (figure 3b). The BoM and ISAC models stand out as very poorly performing during all weeks following week 1. Both either strongly overestimate (BoM) or underestimate (ISAC) the ridge. The JMA model and both versions of the ECMWF fail to simulate the persistence of the observed ridge, and after week 1, its simulated amplitude decreases. Similarly to its realistically simulated trough in the NW Pacific, the ECCC model is consistent in its simulated ridge in Western North America, despite a slight positive bias in week

2. All other models diverge from observations in weeks 2-3, with a weaker than observed ridge.

Figure 3c indicates that all models underestimate the strength of the upper-tropospheric North Atlantic stationary ridge during NDJF. In particular, the CNRM, HMCR, BoM and CMA models poorly simulate the ridge at all lags following week 1, and although other models better perform in the North Atlantic region, most of them also simulate a weaker ridge than that observed during all weeks after week 1. The only exception is the NCEP model that succeeds to simulate the ridge with a very

small bias. These results are somewhat consistent with those that have been shown in previous works, such as Garfinkel et al. (2020) and Park and Lee (2021), who demonstrate that GCMs struggle to produce a realistic and strong enough SW in the North Atlantic upper-troposphere.

Overall, the models are more biased over Western North America and over the North Atlantic compared to the Northwest Pacific region beyond week 2-3, and in some cases, even beyond week 1.




**Figure 1.** Anomalies from zonal mean of geopotential height (m) for initializations during NDJF in the NCEP model during: (a) week 1; (d) week 3; (g) week 5; (j) week 6; the middle column (b,e,h,k) is for ERA-I subsampled to match the dates chosen for NCEP; the right column (c,f,i,l) is for the difference between ERA-I and NCEP; black and magenta contours denote the stationary wavenumber-1 and wavenumber-2 components in ERA-I.

**NDJF eddy height field at 500hPa, 50N; week 3**

**Figure 2.** Longitudinal cross-section of 500 hPa eddy geopotential height at $50°$N during NDJF for S2S models (solid lines) and ERA-I (dots) in week 3: (a) S2S models (solid) and ERA-I (dashed); (b) Difference between S2S models and corresponding days in ERA-I; Vertical grey lines denote the Northwestern Pacific, Nothwestern North America, and North Atlantic regions used for Figure 3; Older versions of ECMWF, UKMO and Meteo-France are denoted with a dashed line.





## NDJF initializations

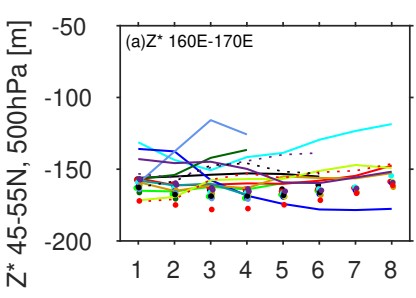
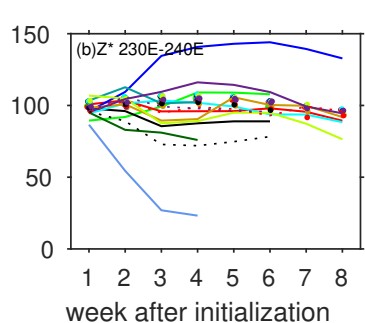
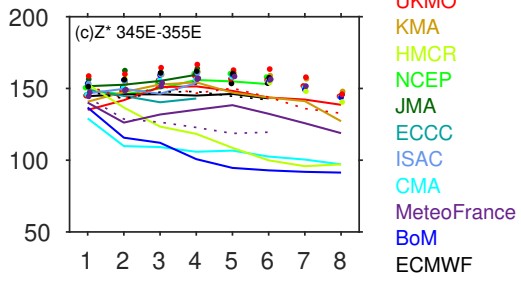

**Figure 3.** Time evolution of simulated 500 hPa stationary eddy height at $45°$N-$55°$N in 3 key regions for all models (solid lines) and corresponding days in ERA-I (dots): (a) Northwestern Pacific ($160°$-$170°$); (b) Western North America ($230°$-$240°$); (c) North Atlantic ($345°$-$355°$); Older versions of ECMWF, UKMO and Meteo-France are denoted with a dashed line.





## 4 Impact of Simulated SWs on Troposphere-Stratosphere Upward Coupling

In the previous section, we focused on the mid-tropospheric SWs. Now, we focus on the stratospheric SWs which are crucial for both a realistically simulated mean circulation in the stratosphere (as we soon show) and temporal variability of the polar
vortex. Figures 4a-i and 5a-i show the observed and modelled NDJF stationary wave structure, both in the troposphere and stratosphere for the NCEP and CMA models, respectively, during week 3 of the integration, when most model simulations start to deviate from observations. In the mid-troposphere, the NCEP simulates a slightly weaker trough over the Northwest Pacific, but the trough extends too far downstream to the East Pacific, as compared to ERA-I (figure 4a-c). In the CMA model, mid-tropospheric biases are more pronounced (figure 5c), with weaker ridges over Western NA and the Northern Euro-Atlantic
region, and a weaker trough in the North Pacific. This dipole pattern, that resembles a negative Pacific North American (PNA) pattern, repeats itself in almost all other models (see supplementary). In the ECMWF-2020 model, the strong biases are mostly evident in western North America where the ridge is too weak (figure S1), while the BoM, HMCR and ISAC models are strongly biased both over the North Pacific and the North Atlantic (figure S5, S7 and S9, respectively). In the stratosphere, biases are weaker in the NCEP model, with the biased negative PNA pattern still present, but decays in the mid-stratosphere at
50 hPa (figure 4). The CMA model, however, does worse in the stratosphere, with a strong underestimation of the predominant wave-1 both in the lower and upper stratosphere. In the CNRM-2019 model, the biases in the stratosphere mostly project onto the stationary wave-2. In the NCEP, ECMWF-2016, UKMO 2016 and 2020 and the KMA models, biases in the stratosphere project onto stationary wave-2, comparable to the biases in the CNRM model, with the KMA and JMA (in the lower strato-sphere) models the least biased (see supplementary). The BoM has a biased wave-2 in the lower stratosphere, while in the
mid-stratosphere it is mostly biased towards wave-1 (figure S5d-i). In the models that struggle to simulate a realistic wave-2 structure, the simulated wave-1, which is predominant in the stratosphere, is still reasonably simulated, therefore, the simulated mean circulation is not expected to be strongly biased. This stands in contrast to the BoM and CMA models that have strongly biased wave-1 in the stratosphere.

To better demonstrate the effect of the regional biases in the troposphere impact the planetary-scale mean NH circulation, which is particularly relevant for the stratospheric mean state, we show in figure 6 different relationships between the S2S mod-els biases in the regional SW field and the amplitude of the stationary wave-1 and wave-2, obtained by Fourier decomposition. Figure 6a contrasts biases in the Northwest Pacific trough with the zonal wavenumber 1 component of the eddy height field at 500hPa. From figure 6a, it is evident that models with a weaker than observed Northwest Pacific trough also have too-weak of a
wave-1 (r=-0.66). Over the North Atlantic, a shallower ridge is associated with a biased too weak wave-1 with a correlation of r=0.55 (figure 6b). Figure 6c contrasts the magnitude of the wave-1 component of eddy height with wave-1 heat flux at 500 hPa, and indicates a strong connection (r=0.91), as expected from Garfinkel et al. (2010). Figure 6d,e show also a strong connection between mid-tropospheric wave-2 biases and regional biases over Western North America (r=0.73) and over Western Eurasia (r=0.67), respectively. Similarly to wave-1, a strong relationship is evident between biases in wave-2 mid-tropospheric height
and biases in the wave-2 component of meridional eddy heat flux (figure 6f). The importance of the Western Eurasian ridge to





upward coupling of wave-2 has been stressed in previous works such as Garfinkel et al. (2019) and Karpechko et al. (2018). In both works, an error in the simulated West Eurasian ridge in S2S models, is linked to a weaker simulated upward propagation of planetary wave activity, and as a result, erroneous stratospheric variability. Figure 6g demonstrates this connection, with biases over Western Eurasia significantly linked to biases in upward propagation of planetary wave-2 activity, indicated by

biases in mid-tropospheric wave-2 meridional eddy heat flux (r=0.65).

The connection of biases in the mid-troposphere to biases in the lower stratospheric planetary wave activity is shown in figures 6h,i, for wave-1 and wave-2, respectively. In both cases, models that are more biased in their simulated wave-1 and wave-2 in the mid-troposphere, are also more biased in their upward coupling of wave-1 (r=0.75) and wave-2 (r=0.78).

In general, the connection between the regional biases and the planetary wave-1 and wave-2 is strong across all regions. In

particular, biases in regional mid-tropospheric SWs are closely related to biases in wave-1 and wave-2 meridional eddy heat flux, which in turn, lead to a biased wave activity that enter the stratosphere.

The wintertime stratospheric polar vortex in the NH is weaker than its Southern Hemisphere counterpart. This is because of the time mean stationary waves, and in particular the wavenumber 1 and 2 components that originate in the troposphere.

Therefore, a key ingredient for a realistic stratospheric mean state is realistic eddy fluxes forced by stationary waves in the troposphere. Figure 7 shows the time evolution of meridional heat flux of wavenumbers 1 and 2 and stratospheric polar vortex, both for the models (solid lines) and the corresponding days in reanalysis (dots) during NDJF. We examine both the mid-tropospheric and lower stratospheric meridional eddy heat flux over the mid-latitudes (40°N-80°N), $\overline{v^*T^*}$, which is used as proxy for upward coupling between the troposphere and the stratosphere.

In the mid-troposphere, wave-1 $\overline{v^*T^*}$ is better simulated by NCEP, UKMO (both versions), ECCC and the KMA models. Both versions of the ECMWF model underestimate the amplitude, particularly following week 2. The CNRM-2019 model accurately simulates $\overline{v^*T^*}$ during the first three weeks, but then underestimates its amplitude, whereas the amplitude in the BoM, ISAC and CMA is poorly simulated and the bias is considerably large throughout the run (figure 7a). For mid-tropospheric wave-2, small biases exist in the NCEP, and both ECMWF and UKMO during the first 4 weeks, while the models that fail to

simulate realistic amplitudes are the CMA, CNRM-2019 and ISAC. (figure 7b). Only the KMA model and to some extent the HMCR model, show consistency, with small biases throughout the run.

In the lower stratosphere, biases in $\overline{v^*T^*}$ are noticeable in most models. For wave-1, both the two versions of the UKMO and ECMWF models decently simulate wave-1 throughout the run, while the KMA model has slightly weaker wave-1 signal than that observed during weeks 2-4, but then it simulates the amplitudes very well. Nevertheless, other models struggle to simulate

a realistic wave-1 throughout the run. The NCEP model largely overestimates the amplitude of wave-1 following week 2, while the CMA, CNRM-2014, ISAC and BoM model considerably underestimate its amplitude after week 1 (figure 7c). Similarly to the mid-troposphere, the CNRM-2019 model starts with a relatively realistic wave-1 amplitude during the first two weeks, but then the simulated amplitude weakens compared to observations. For wave-2 $\overline{v^*T^*}$ shown in figure 7d, both versions of the UKMO model overestimate the amplitude following week 3, whereas both versions of the ECMWF slightly underestimate

the amplitude from week 2. The KMA and CNRM-2019 models are positively biased following week 3 and 1, respectively.



The BoM, ISAC and HMCR have the most poorly simulated wave-2, with too weak amplitudes after week 1, though their negative bias is not as large as in wave-1. The CMA has a decently simulated amplitude of wave-2 during the first 6 weeks of the integration, despite its poorly simulated wave-2 amplitude in the mid-troposphere. In contrast to its overly strong wave-1, the simulated wave-2 in the NCEP model is relatively realistic.

As mentioned earlier, during winter, the presence of stationary waves 1 and 2 in the stratosphere is the reason for a perturbed and weakened vortex, which enables transients to propagate into the stratosphere and perturb the vortex even further. Therefore, a realistic simulated mean state vortex is an important factor for upward wave propagation and vortex daily variability. Figure 7e shows the time evolution of the stratospheric polar vortex, represented as the 10 hPa zonal-mean zonal-wind at $60°$N both in models and reanalysis. Consistent with the large negative bias of wave-1 $\overline{v^*T^*}$ in the lower stratosphere, the CMA and ISAC

models simulate a too strong polar vortex. The NCEP model starts with a realistic vortex strength during the first week, but then diverges from observations and simulates a weaker vortex that becomes even weaker as the run progresses. This is congruent with the substantial positive bias of stationary wave-1 $\overline{v^*T^*}$ in the lower stratosphere shown in figure 7c. The KMA model, and both versions of ECMWF and UKMO models show only small biases, which may be a result a successful simulation of wave-1 in the lower stratosphere by these models. Similarly to the NCEP model, the CNRM-2019 model also simulates a weaker polar

vortex following the first week, but in contrast, this is possibly more due to the positive bias in wave-2 shown in figure 7d. The CNRM-2014, however, is considerably biased towards a stronger vortex than that observed. Interestingly, the ECCC is unable to simulate a realistic polar vortex after the first week, despite its decently simulated waves 1-2 both in the mid-troposphere and lower stratosphere.

Overall, the CMA, HMCR, ISAC and BoM models, i.e, those with a low resolution stratosphere (the latter three are also

low-top within the stratosphere), struggle to simulate realistic $\overline{v^*T^*}$, particularly in the stratosphere, which may be a result of poorly represented upward coupling. This, in turn, affects the stratospheric mean state zonal circulation, and as a consequence, temporal variability is too weak, as demonstrated in previous works (Domeisen et al., 2020b; Schwartz and Garfinkel, 2020). Despite its high model-top and stratospheric resolution, the NCEP and CNRM-2019 models have a too weak vortex strength during winter, which is a result of too strong wave-1 and wave-2 respectively, signal in the lower stratosphere.




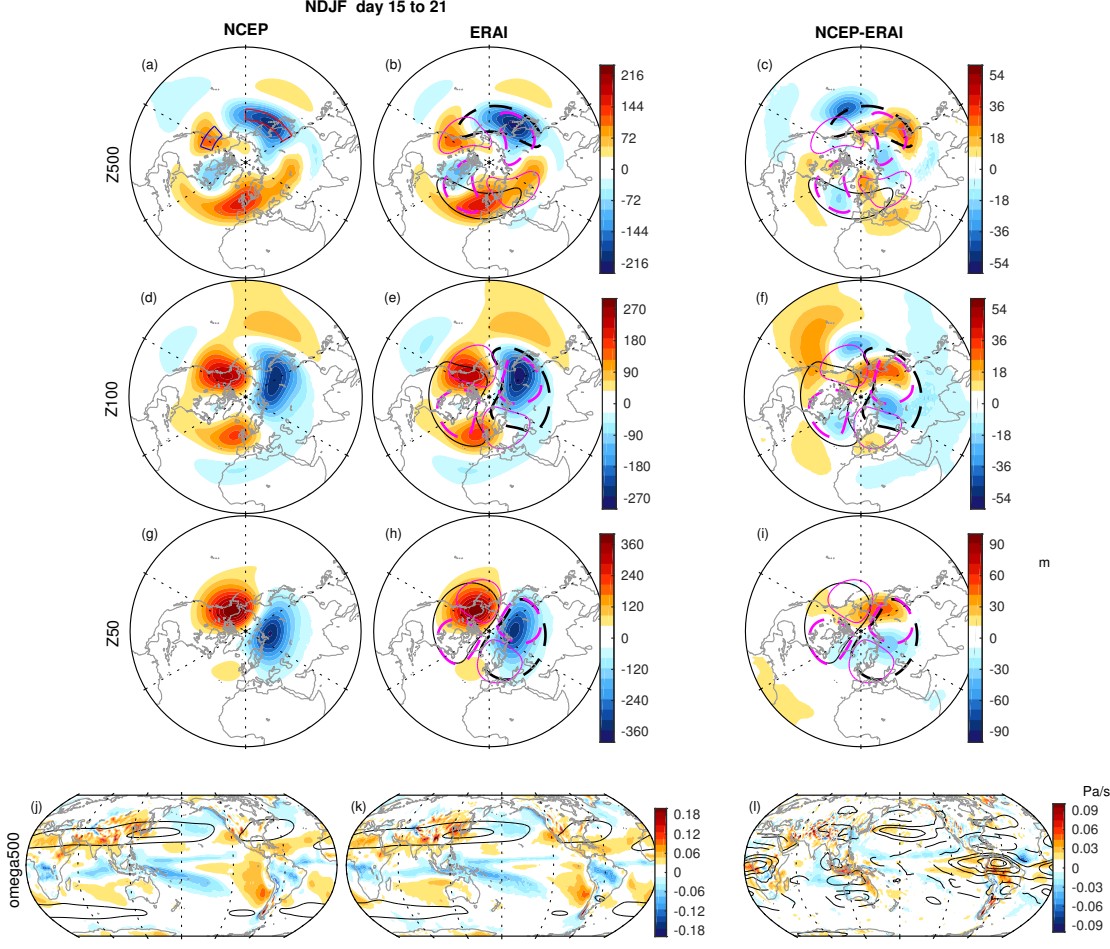

**Figure 4.** Top: anomalies from zonal mean of geopotential height (m) during week 3 for initializations during NDJF in the NCEP model at (a) 500 hPa (d) 100 hPa; (g) 50 hPa; the middle column (b,e,h) is for ERA-I to subsampled to match the dates chosen for NCEP; the right column (c,f,i) is for is for the difference between ERA-I and NCEP; black and magenta contours denote the stationary wavenumber-1 and wavenumber-2 components in ERA-I; Bottom: $\omega$ (color) and zonal wind (contours) during week 3 for initializations during NDJF: (j) in the NCEP model; (k) for ERA-I to subsampled to match the dates chosen for NCEP; (i) the difference between ERA-I and NCEP.



**Figure 5.** Same as figure 4 but for the CMA model.

## relationship among biases, week 3 NDJF

**Figure 6.** Summary of relationship among the S2S models biases during NDJF (the different models are represented by colors similarly to the legend of figure 3);

Top row: Z at 50°N wavenumber-1 and (a) 500 hPa Z* over the Northwest Pacific (160°E-170°E); (b) 500 hPa Z* over the North Atlantic (345°E-355°E); (c) 500 hPa mid-latitude (40°N-80°N) wavenumber-1 v*T*;

Middle row: Z at 50°N wavenumber-2 and (d) 500 hPa Z* over the Western North America (230°E-240°E) ; (e) 500 hPa Z* over Western Eurasia (35°E-55°E); (f) 500 hPa mid-latitude (40°N-80°N) wavenumber-2 v*T*;

Bottom row: (g) 500 hPa mid-latitude (40°N-80°N) wavenumber-2 v*T* and 500 hPa Z* over Western Eurasia (35°E-55°E); (h) 100 hPa mid-latitude (40°N-80°N) wavenumber-1 v*T* and 500 hPa mid-latitude (40°N-80°N) wavenumber-1 v*T*; (i) 100 hPa mid-latitude (40°N-80°N) wavenumber-2 v*T* and 500 hPa mid-latitude (40°N-80°N) wavenumber-2 v*T*;

r denotes the correlation between indices; Dots are for older versions of ECMWF, UKMO and CNRM; All correlations, except for that of panel (a), are significant at the 95% level using a two-tailed Student-t test.



**Figure 7.** Time evolution of all integrations during November-December-January-February for S2S models (solid lines) and corresponding days from ERA-I (dots); older versions are denoted by a dashed line: (a) Mid-latitudes wave-1 $\overline{v^*T^*}$ at 500 hPa; (b) Mid-latitudes wave-2 $\overline{v^*T^*}$ at 500 hPa; (c) Mid-latitudes wave-1 $\overline{v^*T^*}$ at 100 hPa; (d) Mid-latitudes wave-2 $\overline{v^*T^*}$ at 100 hPa; (e) Zonal-mean zonal wind at 10 hPa at 60°N





## 5 Possible Sources for the Biases

As noted in section 1, the biases in the stationary wave pattern, both in the troposphere and the stratosphere, may be a result of a poorly represented time-mean tropical convection. More specifically, deep convection in the Tropical Western Pacific may be one of the drivers that directly (Hoskins and Karoly, 1981; Jin and Hoskins, 1995) and indirectly (Simmons et al., 1983) forms

the PNA pattern. Figures 4 and 5 show the time-mean pressure velocity, $\omega$, and zonal winds during week 3 of the integration in the NCEP and CMA models (panel j), the corresponding days in ERA-I (panel k) and the bias (panel l). In the NCEP model, tropical convection is weaker over the Maritime Continent region compared to ERA-I, but too strong in the Tropical Central and East Pacific. This might contribute to the too weak low, and a too strong downstream extension of the low to the Central and East Pacific that was evident in figure 4. Consistent with this, the Takaya and Nakamura (2001) northward directed flux

is too weak in the West Pacific and too strong in the East Pacific (not shown). In the CMA model, the tropical convection along the South Pacific convergence Zone (SPCZ) is stronger than that observed, but even more evident, are the biases in the mean zonal flow that extend from the Eastern Pacific far into the North Atlantic and may contribute to errors in Rossby wave generation and propagation. The positive bias in $\omega$ over the Maritime continent and West Pacific Warm Pool is evident in all other models, with the NCEP, ECMWF-2016 UKMO-2016, BoM, KMA and CNRM-2019 simulating a too-strong SPCZ.

Next, we examine the regional biases in the tropics that possibly have an impact on the extratropical mean state. Figure 8 shows a longitudinal cross-section of 500 hPa $\omega$, averaged between $15°$S-$10°$N), during week 3 of integration in the models (solid line) and the corresponding days in ERA-I (dots) in panel a, the bias of the models is in panel b.

From panel a, there are three main convection centers along the tropics during NDJF - over Central Africa, the Maritime Continent/West Pacific and the Amazon. Previous works have shown that convection source in these regions can impact the

extratropical North Pacific and North Atlantic (Hoskins and Ambrizzi, 1993; Jin and Hoskins, 1995). Panel b shows that biases in the models, in large, are confined to these regions, and extend to the Eastern Pacific as well. The performance of the models in simulating realistic mean tropical convection is not the goal of this paper, however the divergent wind and subtropical convergence and downwelling in the Northern Hemisphere associated with this tropical upwelling can help seed Rossby waves (Sardeshmukh and Hoskins, 1988). Therefore, we focus now on how the biases in three regions within the tropics and subtropics

are connected to the simulation of SWs in the key regions in the extratropics.

Figure 9 summarises the relationships between NDJF 500 hPa $\omega$ biases in the subtropical Western Pacific, tropical Eastern Pacific and the Caribbean, and the extratropical mid-tropospheric stationary height biases in Northwest Pacific, Western North America and North Atlantic, respectively, during week 3 of integration. Note that following Sardeshmukh and Hoskins (1988), Rossby wave sources (RWSs) are formed in regions of downwelling (and upper tropospheric convergence). Therefore, we

examine subtropical regions where downwelling is considerably strong, as a result of tropical convection in the deep tropics. One exception is the deep tropical Eastern Pacific, which is located in the descending branch of the Walker Cell, therefore is closer to the equator.

Figure 9a indicates that a shallower trough over the mid-troposphere Northwest Pacific is associated with a weaker downwelling than that observed in the subtropical Western Pacific (r=-0.58). Over the Eastern Pacific, enhanced descending motion is





positively correlated with a stronger ridge over Western North America (r=0.56). Similarly, too-weak vertical motions near the Caribbean, are associated with Northwestern Atlantic ridge (r=0.66). In general, figure 9 shows that biases over lower latitudes have a direct impact on the simulated SWs in the extratropics.

In summary, biases emanating from erroneous simulation of tropical convection in the models, are associated with the strength of the extratropical NH stationary SWs, and result in a biased response over the Pacific and Atlantic mid-upper troposphere.



**Figure 8.** longitudinal cross-section of NDJF 500 hPa pressure-velocity ($\omega$) averaged between (15°S-10°N) during week 3 of integration for: (a) S2S models (solid lines) and corresponding days in ERA-I (dots); (b) the difference between S2S models and the corresponding days in ERA-I; Newer versions of ECMWF, UKMO and CNRM are denoted by thin lines.



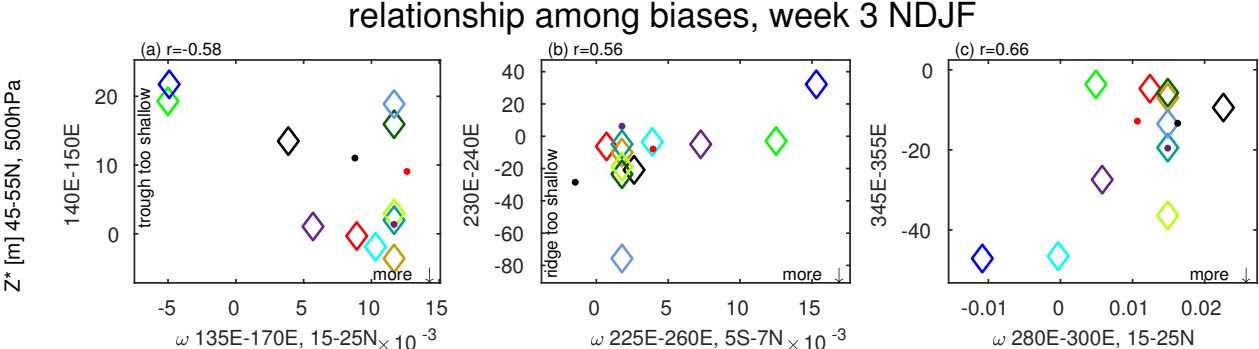

**Figure 9.** Summary of relationship among biases of mid-latitude (45°N-55°N) regional biases of 500 hPa Z* over: (a) The Northwest Pacific (160°E-170°E) and ω over the subtropical Western Pacific (135°E-170°E, 15°N-25°N); (b) Western North America (230°E-240°E) and 500 hPa ω over the Tropical Eastern Pacific (225°E-260°E, 5°S-7°N); (c) The North Atlantic (345°E-355°E) and ω over the Caribbean (280°E-300°E, 15°N-25°N); r denotes the correlation between indices; Dots are for older versions of ECMWF, UKMO and CNRM; All correlations are significant at the 95% level.





## 6 Conclusions

The Northern hemisphere SWs play a major role in regulating the equator-to-pole temperature gradient, as well as modulating the storm tracks and large-scale circulation both in the troposphere and stratosphere. In particular, realistically simulated SWs by global-circulation models used both for future climate projections and medium-range forecasts is crucial for accurate weather and climate surface predictions, which have significant and potentially life-saving, societal and economic implications. In this work, we investigated the how well tropospheric and stratospheric planetary SWs are simulated in 11 operational sub-seasonal models during NDJF. We showed that biases in the troposphere evolve with lead time, and are pronounced already by week 2-3. We showed that nearly all models have like-signed biases over the North Pacific, Western North America and the North Atlantic regions, regions where the amplitude of SWs peaks, and nearly all models simulate too weak a SW in these regions. More specifically, it has been shown that the biases in the North Pacific sector are already evident from week 1, both in amplitude and phase. However, they are smaller compared to those over Western North America and North Atlantic regions; specifically some models suffer from biases of 50% or more in the Northwest North American and North Atlantic ridge.

We also considered SWs biases in the context of troposphere-stratosphere upward coupling, with a particular emphasis on wave-1 and wave-2 which are mostly relevant to upward propagation of waves into the stratosphere. In the troposphere, the models vary in their ability to simulate wave-1 meridional eddy heat flux, while wave-2 is at least decently simulated by most of them, with the KMA model the most realistic in simulated wave-1 and wave-2. Next, we showed that a connection exists between regional biases of SWs in the troposphere and upward propagation of planetary waves, further stressing the importance of realistically simulated SWs in key regions in the mid-latitude troposphere.

In the stratosphere, it is demonstrated that the models with finer vertical resolution in the stratosphere and high model-top generally do a better job in simulating the stratospheric mean state, agreeing with previous works. Interestingly, models with less a complex stratosphere have a more biased wave-1, while those with a finer resolution are more biased in their simulated wave-2. It is not clear if this difference might be a result of small sample size, or alternately a systematic effect of stratosphere complexity in the models. Nevertheless, a more complex stratosphere is only one ingredient for a realistically mean-state stratosphere as demonstrated in the case of the NCEP model, which is positively biased in its wave-1 meridional heat flux in the lower stratosphere, and as a result, its simulated time-mean polar vortex is too weak. This is also shown in Butler et al. (2016) who showed that of the models that participated in the Climate System Historical Forecast Project with a higher complexity stratosphere are not necessarily better skilled in forecasting seasonal sea-level pressure over the North Pole regions in winters when no SST anomalies are present in the tropical Pacific.

Therefore, further work that integrates both the models' mean state and idealized modeling has to be done in order to better identify sources for SWs biases in the S2S models.

Finally, we also considered whether biases in the tropics and subtropics may play a role in the simulated tropospheric mid-latitude SW field in the models. Large biases in the models are evident in oceanic basins immediately to the south and west of the strongest features of the SW pattern, namely in the the Western Pacific, Eastern Pacific and Caribbean. We examined whether regional biases in SWs are related to a biased subtropical (and tropical in the case of the Eastern Pacific) down-





welling, and found a significant connection in the models between a biased downwelling in the subtropical Western Pacific
and Caribbean and biases in the Northwest Pacific trough and Northwest Atlantic. In addition, a significant relationship exists
between downwelling in the tropical Eastern Pacific, associated with the descending branch of the Walker Cell, and the Western
North America ridge.

Our results demonstrate the importance of accurately simulated SWs for both the troposphere and stratosphere mean state in
the models. Most subseasonal models show little consistency in their simulated SWs at lead times longer than 1-2 weeks, which
may enhance forecast errors on subseasonal timescales, particularly those induced by the stratospheric variability.

Overall, it is shown that a well-simulated SWs field in the troposphere is at least a necessary ingredient for a realistically
simulated stratospheric mean circulation, which in turn, enables more accurate temporal variability and surface predictability
on subseasonal timescales.

## 7   Acknowledgements

C. I. G., C. S. and W.C. are supported by the ISF-NSFC joint research program (grant No.3259/19), D.D and P. Y. are supported
by the Swiss National Science Foundation through projects PP00P2_170523 and PP00P2_198896. This work is based on S2S
data. S2S is a joint initiative of the World Weather Research Programme (WWRP) and the World Climate Research Programme
(WCRP). The original S2S database is hosted at ECMWF as an extension of the TIGGE database and can be downloaded
from the ECMWF server http://apps.ecmwf.int/datasets/data/s2s/levtype=sfc/type=cf/. We sincerely thank Ian P. White for the
meaningful discussion throughout the work.

Correspondence and requests for data should be addressed to C. I. G. (email: chaim.garfinkel@mail.huji.ac.il).





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
