# Peer review of "Stationary Wave Biases and Their Effect on Upward Troposphere -Stratosphere Coupling in Sub-seasonal Prediction Models"

_Weather and Climate Dynamics, 2021_

## Author Comment (AC1)

[Figure]

**Figure R1. Left column, from top to bottom: week 1 200 hPa Rossby wave source, 200 hPa wind divergence, 500 hPa ω during NDJF; right column: same as left column but for the NCEP model**

---

## Author Comment (AC2)

relationship among biases, week 3 NDJF

---

## Author Response (AR1)

**Response to Reviewer #1:**

Line 96-98: Would you please be more specific about how the stationary waves are calculated?

Is the time mean geopotential height that is removed while calculating the stationary wave only a one-week average? Thank you. We have added to the text 'weekly mean'.

Is the climatological (multi-year) zonal mean geopotential height removed to calculate the stationary wave? Yes, this is mentioned now: "We define the stationary waves by first computing the weekly mean geopotential height over intializations during November-December-January-February (NDJF) for each model, then compute the climatology for each week, and finally subtract off the zonal mean height at each latitude."

Is a November to February temporal mean removed to calculate the stationary wave? If so, this definition of the stationary wave may not control for annual cycle variability. The stationary wave structure does evolve throughout the winter.

We use the mean for each week, and so we take into account the annual cycle in stationary waves.

Line 110: ERA-Interim is introduced to the reader on this line. I think a line should be added to the Methods section stating that ERA-Interim will be used as the "truth" that the hindcasts will be compared to. Please add a citation for the reanalysis as well. This has been added to the text.

Figure 1: This figure is nice. In its caption, please state what the contour intervals are for the wave-1 and wave-2 contours. Contour intervals have been added to the caption

Figure 1: On panels (i) and (l), the filled contours are not filled where the anomalies are lower than -60 meters. I have come across this as well while plotting. If you are using python and matplotlib, the …extend = 'both'… part of the code below will fill these contours:

```
contour = m.contourf(x, y, vort,
latlon=True,cmap=cmap1,extend='both',levels=levs1,vmin=vmin,vmax=vmax)
```

Thank you, this is fixed now.

Figure 2: Please consider changing your contour colors to something that is more inclusive to people who are colorblind. This article provides guidance: https://www.nature.com/articles/d41586-021-02696-z Thank you, this has been done.

Figure 2a: ERA-I is shown with dashes. Why are there multiple dash contours? Is this ERA-I over different time periods? If yes, please state this in the Figure caption.

The dashed lines were for older versions of the ECMWF, UKMO and Meteo France models, while dots were reanalysis to match the dates for each model.

For simplicity, we decided to remove the old model versions in the revised version of this figure, and similar spaghetti plots. The caption has been revised accordingly.

Line 129: Please replace "observations" with "reanalysis." Replaced

Line 125 – 126: I like that the focus is on these three key regions. Figure 2b gives the impression that there is considerable variability amongst the modeled stationary waves at 200E also. Do you have any hypotheses on why this would happen? Yes, this is due to a more zonally confined NP trough in some models vs others. However this region is not near the maxima/minima of the pattern, and for 2b we prefer the minima.

When we consider the link between convection and North Pacific biases in Figure 9, we now focus on this region (195°E-215°E) instead of 160°E-170°E.

Figure 4: Typo in second line of Figure 4 caption. Thank you, fixed

Figure 4: Please list the contour intervals for the stationary wave and for the zonal wind.

 The contour intervals have been added to the caption.

Line 165 – 166: Is Figure 5c being referenced here? I cannot make out a PNA signal in panel 5c. This is indeed not a classic PNA pattern as it extends into the subtropics. This sentence has been removed from the text.

Line 180: There is a missing word in this sentence. Thank you, it is fixed now.

Line 191: Does either of these studies provide a physical explanation for what causes the biased ridge? If so, please add one line on this. It is difficult to infer a physical mechanism from analyzing the models, as we cannot control the model settings and due to limited data

availability at different pressure levels. For that, an idealized modeling work has to be performed, and this is a work in progress.

Line 284 – 285: Should we expect that the convection over the eastern Pacific is associated with the western North America ridge? The impression I have from Garfinkel et al. (2020) is that the ridge forms due to the nonlinear interactions amongst the "building blocks" that their study focuses on, not tropical convection. To the extent that models represent large-scale topography and land-sea contrast, they already have these two building blocks. Specifically, biases in land-sea contrast would have a rather obvious and immediate impact on surface temperature, winds, and moisture availability (subsequently precip), so we assume models are handling it as well as they can for reasons unrelated to stationary waves. Higher resolution helps resolve topography, but from contrasting the T42 vs T85 experiments in Garfinkel et al 2020 the added value in increasing resolution is not large. Hence our focus in this paper is on the role of tropical convection which varies qualitatively among the models, though we can't rule out other sources of bias.

More generally, we have lowered the degree of confidence implied when we discuss the role of convection for stationary wave biases.

Figure 9a: The connection between tropical convection, subtropical descent, and the East Asia trough is plausible. I feel like the investigation of if tropical convection/subtropical downwelling impacts the stationary wave pattern could be a little more thorough. I think this study could be improved by further investigating the sources of the stationary wave biases.

We have added a figure to the supplemental material showing correlations of these SW features with omega globally. Overall, we agree that demonstrating convincingly **the source** of a stationary bias from model output is a difficult task even if we had full access to model output rather than what is archived by the S2S project.

Thus, we have lowered the degree of confidence implied when we discuss the role of convection. Further, we have work in progress currently using idealized modeling trying to more closely pinpoint how convection and zonal wind biases lead to SW biases. For now, this study aims to identify the stationary wave biases in the models and suggest possible sources that are now further investigated using modeling work.

Have you considered the Rossby wave source? Scaife et al. (2017) did a similar analysis – analyzing the relationship between tropical precipitation and tropics-extratropics Rossby wavetrains. See their Figure 6. Here are some suggested plots: (1) subtropical Rossby

wave source as a function of tropical omega; (2) North Pacific trough bias as a function of subtropical Rossby wave source bias; (3) Rossby wave source maps; (4) North Pacific trough bias as a function of 200 hPa subtropical velocity potential bias.

Scaife, A. A., Comer, R. E., Dunstone, N. J., Knight, J. R., Smith, D. M., MacLachlan, C., ... & Slingo, J. (2017). Tropical rainfall, Rossby waves and regional winter climate predictions. Quarterly Journal of the Royal Meteorological Society, 143(702), 1-11.

Thank you for this suggestion. We had indeed already computed the RWS for three of the considered models (CMA, NCEP, UKMO), but we found omega at 500hPa more conclusive. Specifically, there exist significant differences in RWS between models, likely related to model biases, and hence, 500hPa omega provides a better illustration that is easier to compare between models. Hence, we have decided not to include the RWS results into the manuscript.

For the reviewer's interest, we have included figures of the RWS for these three models into the reviewer response, see Figure R1.

As an example, the Figure below shows the climatology of RWS at 200hPa, divergence at 200hPa, and omega at 500hPa in week 1 of NCEP as compared to ERA-I, for hindcasts initialized in NDJF. Note that the S2S database only includes the pressure levels 300hPa, 200hPa, and 100hPa, which limits the resolution with which we can compute RWS. The RWS and divergence at 200hPa are noticeably too weak in NCEP as compared to ERAI, even as omega at 500hPa is largely reasonable. Note that this is in week 1, when the initial conditions should still be playing a large role and one would hope that the models are doing a reasonable job.

[Figure]

**Figure R1. Left column, from top to bottom: week 1 200 hPa Rossby wave source, 200 hPa wind divergence, 500 hPa ω during NDJF; right column: same as left column but for the NCEP model**

Our interpretation of this result is that too little of the divergent outflow in NCEP occurs at 200hPa, even though convection is occurring in the correct location with a reasonable mass transport, as the omega 500hPa climatology is reasonable.  More generally, the amplitude of tropical and subtropical omega 500hPa is reasonable in most models, and hence we elect to focus on omega in the paper. Given the low vertical resolution available (and also the lack of diabatic heating) in the S2S archive, it is not possible to cleanly identify biases in the convective profile of each model, though we suspect that such biases exist. Future work should consider this issue in more detail assuming more detailed output data becomes available.

Figure 9a: Schwartz and Garfinkel (2020, JGR, MJO study) showed that there is more eddy heat flux entering the mid-latitude stratosphere 1-3 weeks after MJO phase 6/7, suggesting that there is an anomalous tropics-extratropics wavetrain producing the transient eddy heat flux. The convection center during phase 6/7 is between 140E and 180E. Figure 9a looks at subtropical omega between these same longitudes. Assuming that the subtropical descending branch of the meridional circulation between these longitudes is "feeling" what is taking place in the tropics, to what extent is MJO variability present in Figure 9a? Does the Figure 9a correlation improve by compositing by MJO phase?

The MJO is a transient variability on subseasonal timescales. By averaging over many years of data and many initializations within the NDJF season, variability associated with the MJO is filtered out.

**Response to Reviewer #2:**

1) Use and choice of small regions for bias characterization

The discussion around l125 suggests that the largest biases arise near the peaks and troughs of the observed stationary wave. In comparing Fig. 2a and b don't see this at all. In particular, I worry that focusing the discussion on these quite narrow (10 degree by 10 degree) regions can give quite an incomplete view of the nature of the biases across the S2S models. I worry that Figs. 3, 6, and 9 may be quite sensitive to these choices. At a minimum there should be some demonstration that the inferred connections between biases are not sensitive to these choices, and this should be in the manuscript, not just in the response to reviewers. It would also be very helpful to see maps of intermodel correlations in some cases (more on this below). I also wondered if the analysis might be more powerful if the focus was on amplitude and phase of the leading wavenumber components of the anomalies.

We now consider wider areas of 20 degrees by 20 degrees, and the results remained unchanged. If anything, the correlations are even stronger.

In general the phases of wv1 and wv2 are well captured by essentially all models, however the amplitude is more of a mixed bag. The amplitudes of wv1 and wv2 are already shown on figure 6, which connects these amplitudes to regional biases in Z*.

2) Connection to tropical convection

It is certainly very reasonable to hypothesize that these biases could be related to biases in tropical convection. But I again find the evidence presented to be pretty weak: I am not at all convinced that the first place a modeling group should turn to to correct these errors is the tropical mean convection. In part the correlations are relatively weak. Moreover, this is again based on correlations of very small regions. One way to make this connection more convincing may be to show inter-model correlation maps of omega versus geopotential height biases. This would indicate whether the biases have a teleconnection pattern.

We have made figures of the correlation between omega biases across models and Z biases across models as requested. See supplemental Figure S13 in the revised paper. The results from this figure support the paper. More generally, we have lowered the degree of

confidence implied when we discuss the role of convection for stationary wave biases, as we cannot demonstrate causality.

3) Connection between stratospheric bias and stratospheric resolution

This is a simple request (hopefully), but it takes a lot of effort to determine which symbol in a given plot corresponds to which model, and in particular, which symbol corresponds to a high resolution vs low resolution model. It would help to have a different kind of symbol for models in these categories; in particular this seems more useful than distinguishing model versions from individual models.

Old model versions have been removed from figure 2 and similar spaghetti plots. We also added diamonds to low-top models.

A closely related question: Is the wave two component of low-resolution models in better agreement with observations than those of high-resolution models, or is it just that the wave one biases dominate in these cases?

For wave-2, biases in low-top models are comparable in magnitude to those of high-top models, especially in the troposphere (ISAC is an exception). For wave-1, biases in low-top models are more pronounced, therefore it is indeed wave-1 that dominates the biased mean state in the stratosphere.

4) Importance of outliers

In many of the inter-model correlation plots there are one or two models that are to some extent outliers and in some cases seem to be determining the overall correlation (at a quick glance: Fig 6d,e,g; 9b). Some discussion should be included about the sensitivity of these correlations to such outliers.

In figure 6, the outliers have been removed and the correlation coefficient has increased in most panels. Attached is figure 6 without the outliers:

[Figure]

Note that the outliers in the originally submitted version of figure 9 were for models where we had a bug in the initial calculation. This bug has been fixed.

Further questions/comments

1) The authors choose to stratify forecasts by model version in some cases as a result of updates the forecast model over the course of the S2S project. Is there any evidence that these biases depend on model version and not just on sampling errors due to the different time periods? My impression from some single model studies was that the difference was fairly small (I could not easily find a reference for this). In any case, if this is clear it should be presented to justify the extra stratification; if not I would think it better not to stratify the results in this way (?)

Older model versions have been removed from figure 2 and other spaghetti plots. We choose to keep them for the correlation plots (figures 6 and 9), but the reviewer is indeed correct that biases are not substantially changed across model generations.

2) Figure 7 is quite interesting in that it suggests some connection between the stationary wave biases and the zonal mean state. One point of clarification - are the heat fluxes from the stationary component alone?

The heat fluxes are computed using daily data, and then we average over many initializations to get the time mean heat flux. So this isn't a true stationary wave heat flux (where one would generally take the time mean v and time mean T). However in the Northern Hemisphere the difference between the time mean of the daily heat flux and the heat flux computed using time mean v and time mean T is small. (See e.g. the ERA-40 atlas, though we have reproduced this result using other reanalyses. In the Southern Hemisphere this is not the case.)

This is important in that it provides a connection between these biases and other mean-state biases that could be of strong importance for accurately capturing the impact of the stratosphere on forecast skill, for instance. There are some interesting relationships - for instance, the heat flux forecasts of JMA seem to be about right, whereas the zonal mean wind speeds seem to systematically decay. Also, heat flux biases in the CMA forecasts are larger than those in the ISAC model, but the zonal mean state of the latter seems to diverge more quickly.

The JMA zonal mean winds at 10hPa decay as in reanalysis, so that agrees with the simulated eddy meridional heat flux in the lower stratosphere. As for the CMA and ISAC, for wave-1 the CMA indeed has larger biases, but for wave-2 the biases in ISAC are larger. In fact, ISAC is biased in both wave-1 and wave-2, so it somewhat agrees with its biased 10hPa zonal mean winds. For ECCC on the other hand, there doesn't seem to be a relationship between heat flux biases and U10hPa60N biases

Can the authors comment on the relative role of dynamical and radiative processes in determining the mean bias?

Given the present work, we can only comment on dynamical processes that may contribute to the mean bias. The SNAP subproject of SPARC is currently organizing a comprehensive overview of biases in the stratosphere in the S2S models, and this will include a discussion of the relative role of dynamical vs radiative processes. We have added "Radiative

processes can also contribute to mean-state biases in the stratosphere, and future work should consider the relative role of radiative vs. dynamical processes for mean-state biases."

3) Can the authors comment in the consequences of these biases? Do they correlate with forecast skill in any way? We added 1-2 sentences to the conclusions regarding predictability skill. However, this is not the focus of this work, and is discussed in Domeisen et al. 2020a, and will be further analyzed as part of the SNAP papers on stratospheric biases in S2S models.

Domeisen, D. I. V., Butler, A. H., Charlton-Perez, A. J., Ayarzagüena, B., Baldwin, M. P., Dunn-Sigouin, E., et al (2020). The role of the stratosphere in subseasonal to seasonal prediction: 1. Predictability of the stratosphere. *Journal of Geophysical Research: Atmospheres*, 125, e2019JD030920. https://doi.org/10.1029/2019JD030920

---

## Author Response (AR2)

Stationary wave biases and their effect on upward troposphere-stratosphere coupling in sub-seasonal prediction models - Schwartz et al. 2022

The submitted manuscript evaluates the representation of the stationary wave in the troposphere and the stratosphere during boreal winter in 11 subseasonal forecast models. The stationary wave in all models exhibits wave-1 and wave-2 scale biases in the circulation. By evaluating the relationship between the eddy heat flux and the eddy height amplitudes in the troposphere and the stratosphere, the authors show that stationary wave biases influence stratosphere-troposphere coupling. These biases are more pronounced in models with lower model tops. In an exploratory fashion, tropical convection is considered as a source of the extratropical stationary wave biases. The content of this manuscript will be interesting to readers of Weather and Climate Dynamics.

I appreciate that the authors answered all of my questions in the last review while including additional plots to clarify questions posed during the review. With added specificity in the methods section, figure captions, and with the interpretation of the results, the manuscript feels more polished and is nearing completion in my opinion. However, I think more work is needed. Below will follow a list of a few minor comments and one potentially major comment.

I ask that the authors make a major consideration as it may influence the interpretation of the relationship between the stationary wave and tropical convection. On Figures 4l and 5l, the biases in the wind field are pronounced. In fact, the biases in the NCEP and CMA wind fields appear to be fixed in position and planetary scale (predominantly wave-1). The apparent wave-1 signature in the wind biases is apparent in the winter northern hemisphere, but less so in the southern hemisphere, which will less readily support planetary scale undulations in the circulation during summer. These results are not mentioned in detail (lines 258-260) in the manuscript yet, but it is possible that they should be.

The biases in the zonal wind field on Figures 4l and 5l bear a strong resemblance to the stationary wave itself. The anomalies give the impression of contiguous strips of vorticity rotating cyclonically and anticyclonically between the extratropics and the tropics. In the stratosphere, the stationary wave propagates into the QBO westerlies eliciting a similar tropical wind response (see Hamitlon et al. 2004; Elsbury et al. 2021; Sakazaki and Hamilton 2021). I have not seen this signal in the troposphere before so perhaps Figures 4l and 5l are quite noteworthy. Consider relating the zonally averaged upper tropospheric meridional Eliassen Palm Flux to the tropical wind anomalies in a correlation analysis.

Reviewer 1 asked if it is possible that the stationary wave is causing the tropical convection anomalies rather than the other way around, which is the current interpretation. Reviewer 1's request was not answered and I am not convinced the muddied stationary wave representation is not influencing the tropical ascent/descent. The zonal wind anomalies over the Atlantic in CMA exceed 10 m/s. That is a huge signal, which is not discussed.

The zonal wind anomalies are quite large in CMA (Fig. 5l) compared to NCEP (Fig. 4l), which may stem from the huge underestimation of wave-1 in CMA. Conversely, in NCEP, the zonal wind anomalies are less pronounced because the bias in extratropical wave-1 is weaker, hence the effect of the stationary wave on the tropical circulation is more modest.

Following the above hypothesis, there appears to be a general correlation between westerly tropical zonal wind anomalies and positive tropical omega anomalies. Perhaps anomalous positive du/dz in the upper tropospheric tropics is disruptive to vertical ascent? In an opposite sense, there is a decent amount of evidence in the literature finding a relationship between lower stratospheric QBO easterlies and enhanced convection. If the planetary scale tropical zonal wind anomalies are in fact tied to the underlying omega anomalies, a consideration for this manuscript and subsequent studies is whether or

not poor representation of the stationary wave may force biases in tropical circulation and tropical convection, which could subsequently impact tropics-to-extratropics planetary wave propagation. I think the possibility that the stationary wave is modulating the tropical circulation and thereafter, tropical convection, has to be considered before going forward.

Thank you for your detailed comment. Although it is possible that biases in the extratropical stationary waves structure lead to biases in tropical convection, we do not find this behavior to be an important factor in the models. To better demonstrate it, we attach here the time evolution by reforecast week of omega in the three regions discussed in the main text. It is apparent that the biases in omega are evident already in week 1, despite the biases in stationary height in the extratropics still relatively small after the first week (see figure 3).

[Figure]

Specific comments

Figure 5: Consider mentioning that CMA has a "positive-NAO" (North Atlantic Oscillation) like eddy height anomaly, which may be associated with the stronger than average polar vortex (Figure 6). The positive NAO-like bias in figure 5c is shown for week 3. In figure 7e, the vortex strength is not biased during week 1 and slightly positively biased during week 2. Therefore, the stratospheric contribution does not seem to be very significant for the signal in week 3.

Figure 1: Please double check the value of the climatological stationary wave contours. The climatological amplitude of wave-1 at 50 hPa during December should be a few hundred meters. Figure 1 is for the NDJF stationary wave at 500hPa, so are the black and magenta contours for wavenumbers 1 and 2. The peak is indeed much larger than the contours we show for the climatology, however the caption is correct.

Figure 6: Figure 6d shows the relationship between the northwestern North America climatological ridge bias and the wave-2 eddy heat flux bias. Fig. 6g shows the same, but for the other wave-2 ridge. Do the climatological troughs share similarly strong relationships with the wave-2 eddy heat flux? Yes. This is not shown in the main text, but attached here is the correlation between the bias in the Northwest Pacific stationary trough and the bias in stationary wave-2 amplitude. As we can see, the relationship is strong. Now it is also mentioned in the main text.

[Figure]

Figure 7 and Lines 233 - 237: Why are the polar stratospheric westerlies systematically decaying over time (Fig. 7e)? Nine of the eleven models exhibit this variability. This does not appear to be linked to the time evolution of the stationary waves (e.g., Fig. 7c,d) as the eddy heat flux biases remain constant or decrease over time.

The decay of the vortex strength over the course of the weeks in figure 7e is due to the seasonal cycle. Only the CMA and ISAC show a weak seasonal cycle.

** See Figures 4, S5, S6: Elsbury, Dillon, Yannick Peings, and Gudrun Magnusdottir. "CMIP6 models underestimate the Holton-Tan effect." Geophysical Research Letters (2021): e2021GL094083.

** See Figure 7: Hamilton, K., Hertzog, A., Vial, F., & Stenchikov, G. (2004). Longitudinal variation of the stratospheric quasi-biennial oscillation. Journal of the atmospheric sciences, 61(4), 383-402.

Sakazaki, T., & Hamilton, K. Discovery of Quasi-stationary Equatorial Waves Trapped in Stratospheric QBO Westerly and Easterly Jets. Journal of Geophysical Research: Atmospheres, e2021JD035670.